# Towards the Real-World Analysis of Lumbar Spine Standing Posture in Individuals with Low Back Pain: A Cross-Sectional Observational Study

**DOI:** 10.3390/s25102983

**Published:** 2025-05-09

**Authors:** Elena Muñoz-Gómez, Frederick McClintock, Andrew Callaway, Carol Clark, Raee Alqhtani, Jonathan Williams

**Affiliations:** 1Research Unit in Clinical Biomechanics (UBIC), Department of Physiotherapy, Faculty of Physiotherapy, University of Valencia, 46010 Valencia, Spain; elena.munoz-gomez@uv.es; 2Department of Rehabilitation and Sport Sciences, Bournemouth University, Bournemouth BH8 8GP, UK; fmcclintock@bournemouth.ac.uk (F.M.); acallaway@bournemouth.ac.uk (A.C.); cclark@bournemouth.ac.uk (C.C.); 3College of Applied Medical Science, Najran University, Najran P.O. Box 1988, Saudi Arabia; rsalhyani@nu.edu.sa

**Keywords:** accelerometry, IMU, variability, spine, posture, low back pain, wearable electronic devices

## Abstract

Prolonged periods of standing are linked to low back pain (LBP). Evaluating lumbar spine biomechanics in real-world contexts can provide novel insights into these links. This study aimed to determine if standing behaviour can be quantified, in individuals with LBP, in real-world environments. A three-stage design was used, (i) Verification of a bespoke algorithm characterising lumbar standing behaviour, (ii) Day-long assessment of standing behaviours of individuals with posture-related low back discomfort, and (iii) Case study application to individuals with clinical LBP. Analysis of standing posture across time included variability, fidgeting, and amplitude probability distribution function analysis. The study demonstrated that accelerometers are a valid method for extracting standing posture from everyday activity data. There was a wide variety of postures throughout the day in people with posture-related low back discomfort and people with clinical LBP. Frequency profiles ranged from slightly flexed to slightly extended postures, with skewed bell-shaped distributions common. Postural variability ranged from 3.4° to 7.7°, and fidgeting from 1.0° to 3.0°. This work presents a validated accelerometer-based method to capture, identify, and quantify real-world lumbar standing postures. The distinct characteristics of people with low back discomfort or pain highlight the importance of individualised approaches.

## 1. Introduction

Low back pain (LBP) is a highly prevalent condition affecting, on average, approximately 37% of the adult population [1]. LBP stands as the leading cause of work absenteeism and disability [2], underscoring its significance as a major public health concern. Although the exact cause of LBP often remains unclear, it is often provoked by sustained positions or movements. Prolonged sitting, for example, has been shown to be highly provocative for individuals with LBP [3] and associated with the development of transient LBP in healthy individuals [4,5,6]. As a solution to alleviate sitting-related LBP, many have suggested standing as an alternative [7,8]. However, prolonged standing has not been the panacea hypothesised [9,10]. Prolonged standing has been linked to the development of LBP [11,12] and is associated with the perceived LBP or discomfort comparable to seated postures [13]. Therefore, standing postures can be provocative for individuals with LBP.

Previous investigations exploring whether the lumbar lordosis or posture differs between individuals with LBP and those without have shown mixed results, with greater lordosis in some [14] and reduced lordosis in others [15]. Even systematic reviews report conflicting findings, demonstrating either no difference [16] or decreased lordosis [17]. This may be explained through the different measurement methods, for example, radiology compared to surface. However, these conflicts may just be due to the flawed notion of posture being a single fixed entity. Our previous work on sitting posture demonstrated a range of postures within individuals [18]. Furthermore, these studies do little to determine the relationship between posture and pain, as individuals are often not in pain at the time of testing. To achieve this, ‘painful’ or provocative standing should be investigated. This approach has been adopted previously, where individuals are asked to complete a standing task whilst pain levels are monitored, resulting in individuals being classified as either ‘pain developers’ (PDs) or ‘non-pain developers’ (NPDs). Significant differences in lumbar spine curvature during such prolonged standing tasks have been observed, including greater lumbar lordosis [19,20,21,22], as well as increased pelvic incidence and sacral slope [19] in PDs. In addition, PDs exhibit altered proprioceptive postural control, with greater centre of pressure displacement in both the mediolateral and anteroposterior directions after prolonged standing [23]. Less frequent lumbar fidgeting [22] and reduced automaticity of postural control was also observed, raising attentional demand [24].

Despite these early insights into provocative standing postural behaviour, to date, most studies of the lumbar spine in the standing position have been conducted in controlled environments (laboratories) for up to a maximum of two hours [21,24]. While these have been instrumental in developing an understanding of provocative standing behaviour, the environmental constraints fail to capture the daily task variability and dynamic and varied postures associated with daily life.

Evaluating the biomechanics of the lumbar spine in real-world contexts can provide further understanding of the standing behaviour of the lumbar spine, including how this relates to the provocation of pain. This study pioneers an ecologically valid approach by assessing lumbar posture in real-world contexts over an extended period, allowing for a more comprehensive understanding of standing biomechanics. Therefore, the aim of this study was to determine if standing behaviour can be quantified in individuals with LBP in real-world environments, exploring how this relates to the provocation of pain. To achieve this, we need to first verify our algorithm (stage 1), second, complete testing on individuals with low back discomfort by performing a day-long assessment of lumbar standing posture (stage 2), and finally, apply the methods in a case study model to individuals with clinical LBP (stage 3).

The primary hypothesis of this study is that individuals with low back pain or discomfort exhibit unique and distinct quantifiable real-world standing postural patterns and that these can be visualised and quantified in a meaningful way using wearable accelerometers.

## 2. Materials and Methods

The aim of stage 1 was to verify the functionality of automatically detecting standing periods from a full day of kinematic data; stage 2 was to assess the standing behaviour of people with low back discomfort, and stage 3 was to evaluate the applicability of the above methods in two individuals with LBP.

Informed consent was obtained from all participants involved in the study. Stages were approved by the Ethics Committee of Bournemouth University (stages 1 and 2), and the Ethics Committee of Najran University (stage 3) and conducted according to the guidelines of the Declaration of Helsinki.

### 2.1. Design, Setting, and Participants

All stages followed a cross-sectional, observational study design. Stage 1 was completed using a blinded assessor in a laboratory setting, and stages 2 and 3 were completed in the real world without environmental constraints.

For stage 1, a single participant (male, 80 kg, 1.73 m) volunteered with inclusion criteria of self-declared good health, not seeking any treatment for leg or back pain, and no known musculoskeletal disorders of the back or lower limb.

For stage 2, six participants (3 male and 3 female) were recruited from Bournemouth University and surrounding communities. Participants were required to be in ‘self-declared’ good health but reported low levels of low back discomfort (<10 mm in the Visual Analogue Scale (VAS), a 0 to 100 mm line with no pain at the beginning of the line and the worst pain imaginable at the end of the line) during prolonged postures. Participants seeking treatment for leg or back pain or those with musculoskeletal disorders of the back or lower limbs were excluded.

For stage 3, two individuals with clinical LBP were recruited in this proof-of-concept stage. The participants had a medical diagnosis of LBP, with pain localised between the 12th rib and inferior gluteal folds, symptoms persisting for at least three months, and pain evoked by postures or tasks throughout the day (movement-evoked back pain). Individuals were excluded if they had symptoms below the gluteal folds, including neurological symptoms; had any major trauma or surgery to the spine or lower limb; or had a diagnosis of neurological or systemic pathology. Before the evaluation, sociodemographic data and clinical characteristics of LBP were collected from the participants. Pain intensity, including average and worst pain over the preceding two weeks, was assessed using a VAS [25]. Additionally, fear of pain and movement avoidance were recorded using the Fear-Avoidance Beliefs Questionnaire (FABQ), which measures fear-avoidance beliefs related to work and physical activity [26]. Disability levels were also assessed using the Roland–Morris Disability Questionnaire (RMDQ) [27].

### 2.2. Sensor Placement and Data Processing

The sensor placement protocol has been previously described [18]. Briefly, three inertial measurement units (IMUs) (Movella Xsens Dots, Enschede, The Netherlands) were attached to the skin over the L1 and S2 spinous processes and the lateral aspect of the right thigh, midway between the lateral epicondyle and greater trochanter, over the iliotibial band.

Accelerometer data from the sensors were captured at 15 Hz and processed using a bespoke algorithm in MATLAB (MATLAB 2023a, MathWorks, Natick, MA, USA). This algorithm calculated absolute angles and relative angles (between adjacent sensors) using the ATAN2 function after correcting for the initial orientation of each sensor [18]. Angle calculation methods have been validated previously [28]. Angle calculations were restricted to the sagittal plane, representing spinal and hip flexion-extension movements. A 4th-order low-pass Butterworth filter with a 1 Hz cut-off was applied to all data to remove high-frequency noise [29].

Standing detection was derived from several conditional statements relating to the orientation of the L1, S2, and thigh sensor; the angle of the hip; and the movement of the thigh sensor. The time trace of the lumbar spine, hip, and thigh angle using the three-sensor setup is shown in Figure 1.

Stage 1 testing involved the participant completing a series of sitting and standing tasks, each lasting approximately 30 s. These tasks were designed to replicate different standing postures commonly adopted by individuals, such as standing on one leg, interspersed with periods of sitting, walking, or lying down (Table 1). To ensure objectivity, the outcome assessor, who was responsible for analysing the data in MATLAB using the bespoke algorithm, remained blinded to the specific tasks, their duration, and order until data processing was complete. They were also not present during data collection.

In stage 2, participants initially completed three trials of full spinal range of motion (ROM) into flexion and extension to determine maximal ROM prior to wearing the sensors for the day, while performing their daily activities. The only restriction was to avoid immersion in water during the data collection period. At the end of the day, participants returned to the laboratory to have the sensors removed and the data downloaded. Details of the biomechanical testing protocol have been previously published, along with the acceptability of wearing the sensors for the whole day [18].

In stage 3, participants wore the three sensors as described above for the whole day. Additionally, they were provided with a pain diary in which they were requested to record the following information throughout the day: (i) the time (hour and minutes) when any pain was provoked, (ii) the position or activity they were engaged in when the pain was provoked, (iii) the location of the pain, and (iv) the position or activity that alleviated the pain. At the end of the day, participants returned to the laboratory to return the sensors and their pain diary.

### 2.3. Data Analysis

All data were imported into MATLAB with our bespoke standing detection algorithm applied. The standing detection algorithm uses data from the thigh sensor and the resultant hip angle to determine that the person is not sitting [18], prior to removing walking based on the acceleration of the thigh sensor. In stage 1, the resulting standing windows were then analysed and compared to known standing windows. In stages 2 and 3, standing windows were only retained if they were of a duration greater than 60 s. From these windows, individual standing behaviour could be described, including the posture (represented by the mean lumbar angle for each window), duration, and variability of the posture (represented by the standard deviation of the angle).

To visualise the standing behaviour across the day, a novel histogram was developed based on our previous work [18]. The width of the bars represents the duration of that standing window, the height represents the standing lumbar spine angle (posture), and the error bars represent the variability of posture within that window (the standard deviation). In addition, a summary of the standing posture throughout the day was quantified using the variables presented in Table 2. The postural behaviour of the lumbar spine in standing was further described according to the methods of Dunk and Callaghan [30] and McClintock et al. [18], where an amplitude probability distribution function (APDF) was used to visualise the frequency of use of specific lumbar postures and the cumulative probability of the postures. In addition, for stage 3, the pain diary was digitised and used to verify incidences of pain provocation in relation to lumbar spine posture for each participant.

Due to the focus of this study being on the verification of the algorithms and the determination of whether the method can be applied to individuals with sub-clinical low back discomfort and then to those with clinical LBP, it is exploratory in nature. Therefore, a small sample size was used. No formal inferential statistics were attempted, with the analysis focusing on descriptive comparisons and visual representations of postural signatures, thus limiting the comparability and generalizability of the findings.

## 3. Results

The standing detection (stage 1) algorithm accurately identified all fourteen standing periods in the dataset (Table 1 and Table 3 and Figure 1). The findings indicate that standing was accurately detected regardless of lumbar posture (lordotic or flat back standing, regions 2 and 3), leg positioning (prop standing on one leg, regions 6 and 7), arm positioning (standing with bilateral arm raises, region 10), or the level of movement variability (still or shifting posture, regions 5, 8, and 11).

Within stage 2, six participants (age 29 ± 5.2 years, weight 78.5 ± 17.4 kg, height 178.6 ± 9.6 cm) completed a full day of data collection without adverse effects. The average total time spent standing during the day was 2070 ± 672 s (10.2 ± 4.8% of measured time).

Probability and frequency plots for each participant can be seen in Figure 2, along with the window-by-window analysis for the day. These results visually demonstrate that standing posture is very individual.

The APDF plots demonstrate that four individuals (P2, P4, P5, and P6) have a negatively skewed bell-shaped curve with the peak in slight extension-related postures. This is mirrored by the probability plots, all demonstrating a smooth sigmoid shape. The spread around the peak is indicative of the variety of postures used, where for P5, the spread is very limited. The APDFs for P1 and P3 are quite different and contain their largest peaks in slightly flexed postures. This is particularly evident in P3 with a single dominant posture around 7 degrees of flexion. The probability graphs for these 2 individuals are also different, where a stepped progression is evident, reflecting the transitions from peaks to troughs.

The histogram plots suggest again that P1 and P3 have a dominant propensity for flexion-based postures, with lots of standing windows suggesting the breaking up of prolonged standing with other activities (i.e., walking or sitting). Of note are the relatively small error bars for P1, representing small amounts of postural adjustments or fidgets, quite in contrast to P4, where the error bars are larger. This is confirmed by the day summary statistics in Table 4, where the mean fidget for P4 is over 3 times that of P1. Moreover, P2 has the most variable posture, with a standard deviation of mean postures at 7.7 degrees.

Within stage 4, two female participants with LBP completed the full day of data collection wearing the sensors through their normal workday. The sociodemographic characteristics of the participants are shown in Table 5. The mean age was 41.0 ± 1.4 years, and the average BMI was 28.9 ± 4.9.

The total and average standing window time through the day was 2850 ± 2249 s and 188.3 ± 12.7 s, respectively. Day summary statistics for standing for the two participants are shown in Table 6.

The APDF plots of Pt1 (Figure 3a) have a distribution like that of a normal distribution with a central mean at around 0°. There is a hint of an exception with a relatively frequent second, slightly flexed posture, suggesting a bimodal distribution. In contrast, the APDF plot for Pt2 (Figure 3b) displays a positively skewed bell-shaped curve, with the highest peaks in slightly flexed postures.

The histogram plots suggest that Pt1 exhibits standing periods in both flexed and extended postures. In contrast, Pt2 shows a dominant flexion pattern, characterised by fewer but longer-standing windows. There are large error bars in both histogram plots, reflecting significant postural adjustments or fidgeting. Additionally, Pt1 and Pt2 demonstrate variable postures, with a standard deviation of mean postures around 5 degrees (Table 6). 

Pain was reported in standing at 1.9° of extension for Pt1 and 10.5° of flexion for Pt2 (Figure 3).

## 4. Discussion

The aim of this stepwise study was to determine if lumbar spine posture in standing could be measured in the real world and over a protracted period when interspersed with other activities of daily living, such as sitting. This study demonstrates that information about standing posture can be extracted from data of usual activities, can be presented in novel ways to demonstrate the individuality of a person’s standing behaviour across time, and can be applied to clinical populations, demonstrated here for individuals with LBP.

This study demonstrates several original contributions to the literature. Firstly, it demonstrates that wearable sensors housing accelerometers can collect real world data, and from this, it is possible to extract and describe standing posture across time from other activities of daily living. Standing is a common pain-provocation activity [13]; therefore, the study of this in the real world, across a prolonged timeframe offers new opportunities to understand the relationship between posture and the provocation of pain. Previous studies have utilised a thigh sensor to successfully determine standing [31]. Radtke et al. [31] evaluated two thigh-worn systems for measuring posture and movement in an office environment. Their protocol included 24 pre-defined tasks simulating sitting, standing, walking, and postural transitions in the workplace, but only five involved standing. In contrast, our study included 14 standing tasks, allowing for a wider range of postures to be observed throughout the day. Furthermore, relying on a thigh sensor only cannot provide information about the lumbar spine posture. Other studies have utilised a 2-sensor setup to measure lumbar spine posture [32]; however, due to the absence of a thigh sensor, the interaction between the thigh and pelvis is not known, making the determination of standing challenging [33]. The proposed method, despite requiring 3 sensors, has previously been shown to be acceptable to the end-user [18] and has now been shown to represent a proven way to explore real-world standing posture.

The second contribution afforded by this study is the step change to real world environments and prolonged data capture. Most studies assessing spinal posture in standing are conducted in controlled laboratory settings during quiet standing [34] and for a maximum of two hours [21,24]. With the proposed approach, if an individual were to stand for the whole day, up to 8 h of standing data would be available, depending on the battery life of the sensors. Such extended data collection affords opportunities to provide original data analysis and synthesis, such as the novel methods proposed in this study, providing clinicians for the first time a detailed understanding of standing behaviour.

The present study included various tasks interspersed with standing, allowing for natural ecological variability and validity of measurement. In fact, individuals with LBP often introduce variability in their spinal posture between brief periods of standing to alleviate discomfort [35]. In addition, both posture and standing-related discomfort may vary throughout the day [36]. Indeed, our results showed that there is a great variety of postures in most cases, with P2 being the person with the most variable posture (7.7 degrees), as well as Pt1 and Pt2 with a posture variation of 5 degrees.

Previous studies have assessed the posture of the spine in the standing position in those who report pain through prolonged standing (PDs). However, there is no consensus on a specific provocative lumbar spine posture. While some authors describe a posture of increased lumbar lordosis compared to NPDs [22]. Others have found no significant differences [20,37]. Indeed, the study by Gregory et al. [38] found that all participants tended to increase the degree of flexion in which they stood over time. This could explain the different results obtained in the present study. Of the individuals with postural related low back discomfort, P2, P4, P5, and P6 showed a posture with slight extension, while P1 and P3 had flexed postures. Similarly, in the two patients with LBP, it was observed that Pt1 had a mean of around 0°, varying between flexion and extension positions during the day, while Pt2 had a predominantly flexed position. This is consistent with what was observed in the study by Gombatto et al. [32], in which participants with LBP were classified into two groups. One group moved less frequently into lumbar flexion, whereas the other group moved more frequently into lumbar flexion. Such individualisation is only visible through this approach, exploring the unique behaviour or change in posture across time.

The third contribution offered from this study relates to variability. The current study demonstrates novel approaches to the quantification of variability, which describes both the range of habitual postures and the ‘fidgeting’ within the postures. Current clinical practice may take a snapshot of standing posture, resulting in erroneous conclusions. Furthermore, analysing such information provides knowledge about the individual’s ability to alter the distribution of load on different structures of the spine [39]; for example, small fidgets are unlikely to alter the stress profile across spinal structures [40]. In this regard, less lumbar fidgeting during standing is associated with an increased risk of standing-related LBP [22]. In the present study, differences in lumbar fidgeting were also observed between the participants. For example, among participants with discomfort, P1 exhibited small fidgets, whereas P4 displayed large fidgets. Pt1 and Pt2 showed average fidgets of 2.4 degrees and 2.5 degrees, which is consistent with the literature [8]. Therefore, this data highlights the distinct characteristics of each participant, reinforcing the importance of personalised approaches.

The current work presents a validated accelerometer-based method of capturing, identifying, and quantifying real-world lumbar spine standing postures, removing task or environmental constraints found within the laboratory. The significant development and original contribution described in this work is the ability to provide long-term, real-world monitoring of standing postural ‘signatures’ through the novel visualisation of posture. Moreover, the opportunity afforded to explore variability across time offers a step change in focus regarding clinical populations, challenging the reductionist notion of a single posture. The sensor data allows clinicians to ‘visualise’ not just standing posture information across time but also those postures associated with the provocation of pain. Understanding the personalised link between their posture and pain provides new possibilities for classification or characterisation of LBP disorders. Examples of such might include those associated with prolonged static postures (as would be seen with low variability or fidgeting), those associated with specific postural profiles (as demonstrated through APDF and histograms), or those associated with a specific postural history (as demonstrated through the histograms). Understanding such links leads to the potential for targeted interventions, potentially addressing variability, habits, or postures leading to the prevention of their individual provocation factor.

The work presented here has several limitations. First, it should be acknowledged that this study only incorporated real-world data from 8 people. Such a small sample size precludes any inferential statistical analysis; however, it does enable the presentation and exploration of the ‘dynamic’ standing behaviour of each individual. Caution is advised when extrapolating results to others. Future studies could utilise the proposed methods to explore a larger sample of people with LBP to determine if clusters of postural behaviour can be identified and linked with LBP provocation. The design of this study was specifically to explore the sagittal plane, and therefore, this can be regarded as a second limitation of the current work. It is understood that, in addition to the sagittal plane, LBP is associated with postures out of the sagittal plane [41]. Future studies could also explore other planes of motion utilising the techniques outlined above. Furthermore, expanding the understanding beyond that of kinematics may offer additional insights; for example, exploring muscle properties and activities and how these are related to the provocation of pain [42,43].

## Figures and Tables

**Figure 1 sensors-25-02983-f001:**
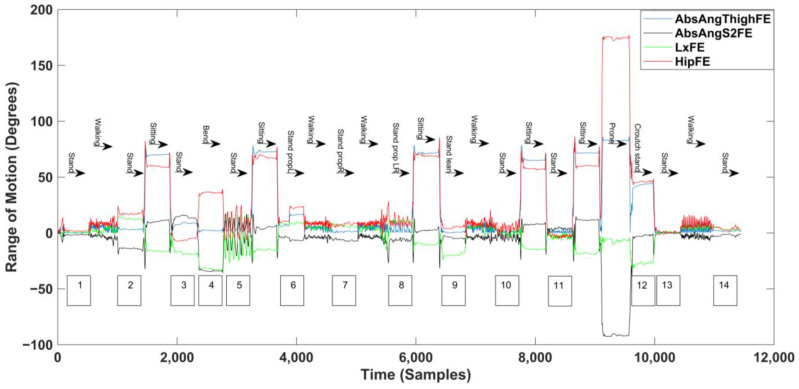
Time-series data of lumbar spine, hip, and thigh angles captured with the three-sensor setup. AbsAngThighFE, Absolute sagittal angle of the thigh; AbsAngS2FE, Absolute sagittal angle of the S2 sensor; LxFE, Lumbar sagittal angle; HipFE, Hip sagittal angle. For meaning of annotation numbers see Table 1.

**Figure 2 sensors-25-02983-f002:**
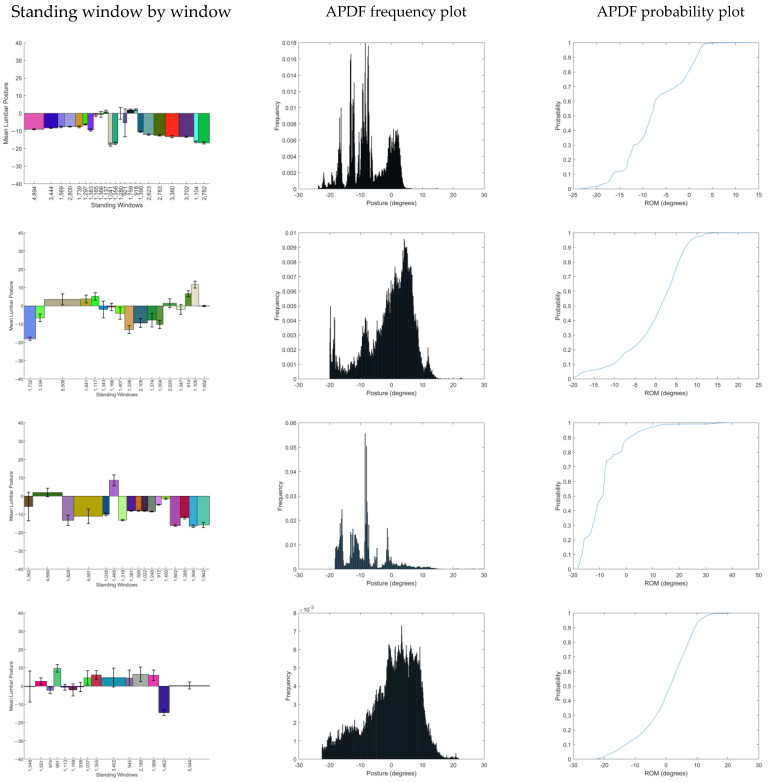
Through-day standing window-by-window analysis (negative posture values represent Flexion, positive posture values represent Extension) and Amplitude Probability Distribution Function for each of the six participants.

**Figure 3 sensors-25-02983-f003:**
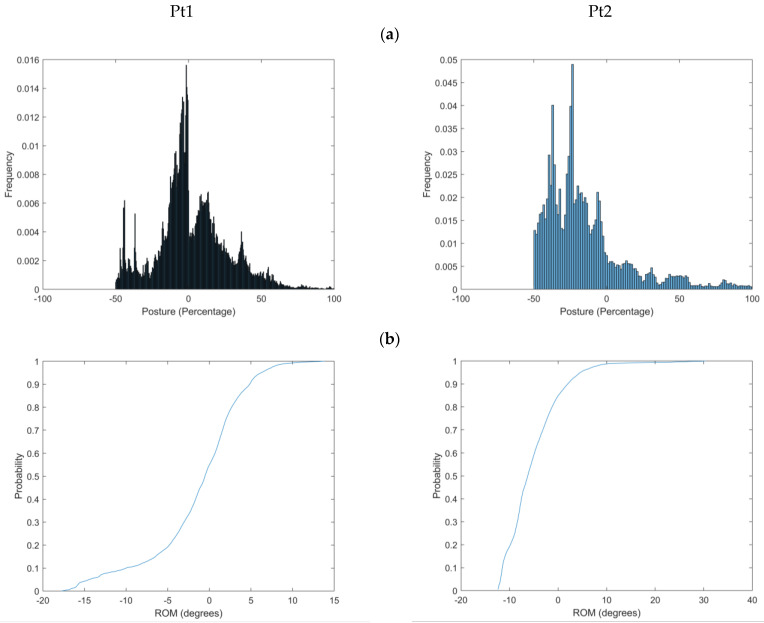
(**a**,**b**), Amplitude probability distribution function (APDF) plots of the two participants with LBP through day postures, (**b**), Cumulative Distribution Frequency plots for two participants with LBP through day postures, (**c**), Through-day window-by-window analysis for each of the two participants with LBP in chronological order. Red denotes a painful standing window.

**Table 1 sensors-25-02983-t001:** Sequence and duration of tasks performed in the blinded verification study.

Approximate Time (s)	Activity	Period of Standing
0–30	Normal standing	1
30–60	Walking	
60–90	Lordotic standing	2
90–120	Sitting	
120–150	Flat back standing	3
150–180	Sustained forward bending	4
180–210	Flat back to lordotic (Sagittally variable posture)	5
210–240	Sitting	
240–270	Prop standing on Left leg	6
270–300	Walking	
300–320	Prop standing on Right leg	7
320–350	Walking	
350–380	Prop standing left to right (Frontally variable posture)	8
380–410	Sitting	
410–440	Sustained half flexion	9
440–470	Walking	
470–500	Standing with bilateral arm raises	10
500–530	Sitting	
530–560	Standing twisting to left and right (transverse variable posture)	11
560–590	Sitting	
590–620	Prone lying	
620–650	Crouched standing	12
650–680	Normal standing	13
680–710	Slow walking	
710–740	Normal standing	14

s: seconds.

**Table 2 sensors-25-02983-t002:** Variables used for throughout-day standing analysis.

Variable	Description
Average standing posture for the day	Derived from the unweighted mean of all the standing windows
Weighted average standing posture for the day	Derived from the weighted mean of all the standing windows
Variability of standing posture across the day	Determined from the standard deviation of the mean posture from each standing period
Average ‘fidgeting’ in standing	Derived from the mean of the standard deviations of the standing windows

**Table 3 sensors-25-02983-t003:** Accelerometer system results in the blinded verification study.

Standing Region	Duration (s)	Mean Lumbar Posture (°) (Extension Positive)	Standard Deviation (°)	Known Standing Position	Standing Identified by Algorithm
1	33	0.6	1.0	Normal standing	Y
2	25	12.2	1.0	Lordotic standing	Y
3	28	−18.0	1.6	Flat back standing	Y
4	26	−30.0	7.9	Sustained flexion	Y
5	28	−3.2	14.6	Flat back to lordotic (Sagittally variable posture)	Y
6	25	8.4	0.7	Prop standing on Left leg	Y
7	23	6.3	0.8	Prop standing on Right leg	Y
8	23	10.4	1.1	Prop standing left to right (Frontally variable posture)	Y
9	25	−19.5	1.5	Sustained half flexion	Y
10	24	0.3	1.8	Standing with bilateral arm raises	Y
11	21	3.2	1.6	Standing twisting to left and right (transverse variable posture)	Y
12	21	−27.3	1.8	Crouched standing	Y
13	28	0.4	0.8	Normal standing	Y
14	33	3.6	1.0	Normal standing	Y

°: degree, s: seconds, Y: yes.

**Table 4 sensors-25-02983-t004:** Day summary statistics for standing for the six participants.

P	No. of Standing Windows	Average Standing Window Duration (s) (SD)	Weighted Average Standing Posture (°)	Unweighted Average Standing Posture (°)	Variability of Standing Posture (°)	Average Fidget When Standing (°)
1	23	132.9 (72.34)	−9.2	−8.2	6.3	1.0
2	21	110.2 (69.8)	−2.0	−2.4	7.7	2.3
3	13	115.3 (75.4)	−7.7	−8.3	6.7	1.5
4	15	113.6 (82.7)	1.7	1.6	5.7	3.2
5	16	164.8 (111.6)	4.1	3.1	4.1	2.4
6	10	121.5 (72.4)	−2.4	−1.1	3.4	2.0

P: participant, s: seconds, No.: number, °: degrees.

**Table 5 sensors-25-02983-t005:** Sociodemographic and LBP clinical characteristics of the participants.

	Pt1	Pt2
Sex	F	F
Age (years)	42	40
Height (cm)	1.7	1.6
Weight (kg)	70	83
BMI (kg/cm^2^)	25.4	32.4
Duration of LBP (months)	7	156
Average pain (VAS)	2.9	3.8
Worst pain (VAS)	4.5	4.2
FABQ Physical activity subscale	3	13
FABQ Work subscale	10	19
RMDQ	6	13

F: female; cm: centimetres; kg: kilograms; VAS: visual analogue scale; FABQ: Fear-avoidance beliefs questionnaire; RMDQ: Roland–Morris disability questionnaire.

**Table 6 sensors-25-02983-t006:** Day summary statistics for standing for the two participants.

P	No. of Standing Windows	Mean Standing Window Duration (s) (SD)	Weighted Mean Standing Posture (°)	Unweighted Mean Standing Posture (°)	Variability of Posture (°)	Average Fidget (°)
Pt1	23	193.13 (196.39)	−0.7	−2.0	5.0	2.4
Pt2	7	179.34 (147.65)	−1.8	−3.2	5.2	2.5

P: participant, No.: number, s: seconds, °: degrees.

## Data Availability

Data can be obtained by request from authors.

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
