# Peer review of "Towards the Real-World Analysis of Lumbar Spine Standing Posture in Individuals with Low Back Pain: A Cross-Sectional Observational Study"

_sensors, 2025, doi:10.3390/s25102983_

Round 1

Reviewer 1 Report

Comments and Suggestions for Authors

Dear authors,

First of all thank you for the invitation to review your study “ Towards the Real-World Analysis of Lumbar Spine Standing  Posture in Individuals with Low Back Pain.” I thank the authors for their efforts in producing this study that aligns with my expertise. Please find some specific comments below.

TITLE:

- I suggest to add A cross-sectional observational study

ABSTRACT:

-I suggest to provide more details for the statistical analysis of the methods.

- I suggest to provide data in the results

-Please adopt MeSh terms as keywords.

- typo error “Keywords: accelerometer, IMU, variability, spine, posture, low back pain” line 29

INTRODUCTION

-The background and rationale part could be better presented. I suggest rewriting it with a description of what is known and of what is missing and emphasise the importance of your study in the field.

- lines 89-92 add it in the method section not introduction

-I would add the hypothesis.

METHODS

-being a cross sectional study, it should be be adherent to the international guidelines (STROBE). I suggest to follow it and to consider all the items. I also expect more details for methods and the completed checklist to be loaded as supplementary.

RESULTS:

- table 3 should be presented as table 2 and table 1, such as with only three lines

-I would divide the results into sections following STROBE, this should confer more structure and could be more related to method section.

DISCUSSION

-I suggest the authors discuss the main findings by comparing them with the existing literature, suggesting implications, and analysing strengths and limitations.

-Implications for practice and research should be developed more in-depth. Future studies should assess muscles activities and proprieties related to LBP with tensiomyography (doi: 10.1080/10669817.2023.2252202) and with electromyography  (doi: 10.1016/j.jelekin.2012.11.010). Please take in to consideration these articles in your discussion.

Author Response

Firstly, may I thank the reviewers for taking the time to review our manuscript and provide helpful comments resulting in an improved manuscript. Second, please see our responses below and in red in the manuscript.  

Review 1

Dear authors,

First of all thank you for the invitation to review your study “ Towards the Real-World Analysis of Lumbar Spine Standing  Posture in Individuals with Low Back Pain.” I thank the authors for their efforts in producing this study that aligns with my expertise. Please find some specific comments below.

TITLE:

- I suggest to add A cross-sectional observational study

Thank you for the suggestion, we have added this.

ABSTRACT:

-I suggest to provide more details for the statistical analysis of the methods.

We agree and have added more detail in the abstract.

- I suggest to provide data in the results

Many thanks, we have added some data in the abstract.

-Please adopt MeSh terms as keywords.

- typo error “Keywords: accelerometer, IMU, variability, spine, posture, low back pain” line 29

Thank you, we have reviewed the terms and where appropriate these have been updated to to align with MeSH.

INTRODUCTION

-The background and rationale part could be better presented. I suggest rewriting it with a description of what is known and of what is missing and emphasise the importance of your study in the field.

We thank you for this suggestion and agree some of the introduction could be improved. We have reviewed and made significant edits.

- lines 89-92 add it in the method section not introduction

This has been moved to the correct section.

-I would add the hypothesis.

A hypothesis has been added at the end of the introduction.

METHODS

-being a cross sectional study, it should be be adherent to the international guidelines (STROBE). I suggest to follow it and to consider all the items. I also expect more details for methods and the completed checklist to be loaded as supplementary.

We have rewritten this version of the manuscript using the strobe guidelines. Furthermore, we include a completed checklist.

RESULTS:

- table 3 should be presented as table 2 and table 1, such as with only three lines

This has been completed; however, the table order has now changed.

-I would divide the results into sections following STROBE, this should confer more structure and could be more related to method section.

This has now been edited to align better to the strobe guidelines.

DISCUSSION

-I suggest the authors discuss the main findings by comparing them with the existing literature, suggesting implications, and analysing strengths and limitations.

-Implications for practice and research should be developed more in-depth. Future studies should assess muscles activities and proprieties related to LBP with tensiomyography (doi: 10.1080/10669817.2023.2252202) and with electromyography  (doi: 10.1016/j.jelekin.2012.11.010). Please take in to consideration these articles in your discussion.

Many thanks for this comment. We have added some additional sections to strengthen the discussion raising the implications particularly.

Reviewer 2 Report

Comments and Suggestions for Authors

Review of “Towards the real-world analysis of lumbar spine standing posture in individuals with low back pain” by Elena Muñoz-Gómez, Frederick McClintock, Andrew Callaway, Carol Clark, Raee Alqhtani and Jonathan Williams

The manuscript explores the feasibility of using wearable IMUs to assess lumbar spine posture during standing in real-world settings, especially in individuals with low back pain (LBP). The study provides a proof-of-concept that real-world posture variability can be captured effectively. However, the current manuscript suffers from several limitations that, in my view, restrict its value in its current form.

Below, I provide a detailed critique with some suggestions:

1. The manuscript is divided into multiple distinct "studies", each with full-length sections for methods and results. This segmented approach creates a fragmented reading experience and inflates the apparent scope of the work. It may also give an impression of trying to artificially amplify the contribution by presenting small-scale efforts as standalone components. It is strongly recommended to consolidate the three components into a more unified narrative.

2. Each component of the study relies on a minimal number of participants:

- Study 1 involves a single healthy subject.

- Study 2 includes only 6 individuals with “self-reported postural discomfort”.

- Study 3 presents just 2 case studies of individuals with chronic LBP.

Such small numbers preclude any generalization or robust statistical analysis. The results are inherently descriptive and exploratory, which should be acknowledged more explicitly throughout the manuscript. The authors refer to these as “studies”, but this framing may be misleading; these are more appropriately viewed as pilot investigations or methodological trials.

3. The analyses are mostly descriptive, with no inferential statistics or hypothesis testing. Given the variability among individuals and the small sample size, this is understandable. However, the manuscript would benefit from a more formal acknowledgment of this limitation, along with a discussion of how future work could incorporate appropriate statistical designs.

4. There is no discussion of how this sensor-based data could be integrated into decision-making or treatment. Furthermore, the pain-provocation data from the case studies are anecdotal and not clearly linked to measurable biomechanical patterns. The time-window histograms seem interesting, but their interpretive value is unclear

5. The authors acknowledge that the analysis is restricted to the sagittal plane. This is a significant limitation, particularly for lumbar spine kinematics, where transverse and frontal plane movements also contribute to discomfort and compensatory behavior.

Author Response

Firstly, may I thank the reviewers for taking the time to review our manuscript and provide helpful comments resulting in an improved manuscript. Second, please see our responses below and in red in the manuscript.  

Review 2:

Review of “Towards the real-world analysis of lumbar spine standing posture in individuals with low back pain” by Elena Muñoz-Gómez, Frederick McClintock, Andrew Callaway, Carol Clark, Raee Alqhtani and Jonathan Williams

The manuscript explores the feasibility of using wearable IMUs to assess lumbar spine posture during standing in real-world settings, especially in individuals with low back pain (LBP). The study provides a proof-of-concept that real-world posture variability can be captured effectively. However, the current manuscript suffers from several limitations that, in my view, restrict its value in its current form.

 Below, I provide a detailed critique with some suggestions:

  1. The manuscript is divided into multiple distinct "studies", each with full-length sections for methods and results. This segmented approach creates a fragmented reading experience and inflates the apparent scope of the work. It may also give an impression of trying to artificially amplify the contribution by presenting small-scale efforts as standalone components. It is strongly recommended to consolidate the three components into a more unified narrative.

We thank you for this suggestion. We have restructured the manuscript as you have recommended.

  1. Each component of the study relies on a minimal number of participants:

- Study 1 involves a single healthy subject.

- Study 2 includes only 6 individuals with “self-reported postural discomfort”.

- Study 3 presents just 2 case studies of individuals with chronic LBP.

Such small numbers preclude any generalization or robust statistical analysis. The results are inherently descriptive and exploratory, which should be acknowledged more explicitly throughout the manuscript. The authors refer to these as “studies”, but this framing may be misleading; these are more appropriately viewed as pilot investigations or methodological trials.

Many thanks for this comment. We have removed the study descriptor and have removed the study-by-study approach as outlined in your earlier comment. Furthermore, we have added a section acknowledging the lack of generalizability.

  1. The analyses are mostly descriptive, with no inferential statistics or hypothesis testing. Given the variability among individuals and the small sample size, this is understandable. However, the manuscript would benefit from a more formal acknowledgment of this limitation, along with a discussion of how future work could incorporate appropriate statistical designs.

Many thanks for this comment. We have added a section acknowledging the lack of generalizability in the methods, as we discuss the absence of statistics. Furthermore, we have added some elements to the discussion. 

  1. There is no discussion of how this sensor-based data could be integrated into decision-making or treatment. Furthermore, the pain-provocation data from the case studies are anecdotal and not clearly linked to measurable biomechanical patterns. The time-window histograms seem interesting, but their interpretive value is unclear

Many thanks for this comment. This has now been addressed in the discussion section.

  1. The authors acknowledge that the analysis is restricted to the sagittal plane. This is a significant limitation, particularly for lumbar spine kinematics, where transverse and frontal plane movements also contribute to discomfort and compensatory behavior.

We agree with this and have strengthened the text around this limitation.

Round 2

Reviewer 1 Report

Comments and Suggestions for Authors

I thank authors for the revision done

Reviewer 2 Report

Comments and Suggestions for Authors

I thank the authors for their careful revisions. The paper is now clear and well-structured. I recommend it for publication.

PS: “8. Discussion” should be renamed “4. Discussion”